# Association between Regional Tissue Oxygenation and Body Temperature in Term and Preterm Infants Born by Caesarean Section

**DOI:** 10.3390/children7110205

**Published:** 2020-10-29

**Authors:** Marlies Bruckner, Lukas P. Mileder, Alisa Richter, Nariae Baik-Schneditz, Bernhard Schwaberger, Corinna Binder-Heschl, Berndt Urlesberger, Gerhard Pichler

**Affiliations:** 1Research Unit for Neonatal Micro- and Macrocirculation, Department of Pediatrics and Adolescent Medicine, Medical University of Graz, 8036 Graz, Austria; marlies.bruckner@medunigraz.at (M.B.); richteralisa92@gmail.com (A.R.); nariae.baik@medunigraz.at (N.B.-S.); bernhard.schwaberger@medunigraz.at (B.S.); corinna.binder@medunigraz.at (C.B.-H.); berndt.urlesberger@medunigraz.at (B.U.); gerhard.pichler@medunigraz.at (G.P.); 2Division of Neonatology, Department of Pediatrics and Adolescent Medicine, Medical University of Graz, 8036 Graz, Austria

**Keywords:** body temperature, hypothermia, hyperthermia, neonates, term, preterm, postnatal transition, oxygenation, tissue oxygenation, near-infrared spectroscopy

## Abstract

Body temperature (BT) management remains a challenge in neonatal intensive care, especially during resuscitation after birth. Our aim is to analyze whether there is an association between the BT and cerebral and peripheral tissue oxygen saturation (crSO_2_/cTOI and prSO_2_), arterial oxygen saturation (SpO_2_), and heart rate (HR). The secondary outcome parameters of five prospective observational studies are analyzed. We include preterm and term neonates born by Caesarean section who received continuous pulse oximetry and near-infrared spectroscopy monitoring during the first 15 min, and a rectal BT measurement once in minute 15 after birth. Four-hundred seventeen term and 169 preterm neonates are included. The BT did not correlate with crSO_2_/cTOI and SpO_2_. The BT correlated with the HR in all neonates (ρ = 0.210, *p* < 0.001) and with prSO_2_ only in preterm neonates (ρ = −0.285, *p* = 0.020). The BT was lower in preterm compared to term infants (36.7 [36.4–37.0] vs. 36.8 [36.6–37.0], *p* = 0.001) and prevalence of hypothermia was higher in preterm neonates (29.5% vs. 12.0%, *p* < 0.001). To conclude, the BT did not correlate with SpO_2_ and crSO_2_/cTOI, however, there was a weak positive correlation between the BT and the HR in the whole cohort and a weak correlation between the BT and prSO_2_ only in preterm infants. Preterm neonates had a statistically lower BT and suffered significantly more often from hypothermia during postnatal transition.

## 1. Introduction

Maintaining optimal body temperature remains a challenge in neonatal intensive care. In 1997, the World Health Organization (WHO) defined the normal range of body temperature in newborn infants between 36.5–37.5 degrees Celsius (°C) [1]. However, Knobel et al. observed that the maximized normal heart rate (HR) observations occurred when the neonates body temperature was 36.8–37.0 °C [2], and Lyu et al. described the lowest rates of adverse neonatal outcomes in preterm infants with body temperatures of 36.5–37.2 °C [3]. Generally, the body temperature of newborn infants varies widely and depends on different variables such as postnatal age, location of measurement, and the method to determine body temperature [4,5,6]. Both hypothermia and hyperthermia are associated with increased mortality and morbidity [3,7]. The incidence of neonatal hypothermia is up to 84% in developing countries [8] and up to 53% in Europe [9], whereby preterm neonates are more often affected compared to term neonates because of differences in body temperature management [10]. Pichler et al. reported that body temperature correlated with peripheral muscle tissue oxygenation, measured by near-infrared spectroscopy (NIRS), in newborn infants who were admitted to the neonatal intensive care unit [11]. Whether body temperature affects both peripheral and cerebral tissue oxygenation in newborn infants during fetal-to-neonatal transition remains unclear.

The primary aim of this study is to analyze whether there is an association between (i) body temperature and cerebral (crSO_2_/cTOI) and peripheral regional tissue oxygen saturation (prSO_2_) and between (ii) body temperature and the HR and arterial oxygen saturation (SpO_2_) in neonates born by Caesarean section during fetal-to-neonatal transition. Our secondary aim is to (iii) describe and compare body temperatures between term and preterm neonates of the same cohort in minute 15 after birth. We hypothesize that body temperature will not affect crSO_2_/cTOI, due to auto-regulative processes. For prSO_2_, we expect a negative correlation with body temperature, due to a consequently increased metabolic rate. Further on, we hypothesize that the HR and SpO_2_ will correlate positively with body temperature.

## 2. Materials and Methods

This study is a post-hoc analysis of secondary outcome parameters of five prospective single-center observational studies that were performed between 2008 and 2016 at the Division of Neonatology, Department of Pediatrics and Adolescent Medicine, Medical University of Graz, Austria. All studies were approved by the Ethics Committee of the Medical University of Graz, Austria (EK-numbers: 19-291ex07/08, 23-302ex10/11, 25-342ex12/13, 25-592ex12/13, and 27-465ex14/15), with written parental consent being obtained prior to study inclusion.

### 2.1. Participants

Term and preterm neonates who received continuous NIRS and pulse oximetry monitoring during the first 15 min after birth as well as body temperature measurements in minute 15 after birth were included. Due to logistical reasons, measurements were performed only in neonates delivered by elective, interactive, or emergency Caesarean section. Exclusion criteria were vaginally delivered newborns and presence of congenital malformations that could potentially affect cardiorespiratory or neurological function.

### 2.2. Postnatal Stabilization and Temperature Management

Cord clamping was performed within 30 s after birth. After that, all neonates were immediately placed in supine position under the resuscitation table’s (CosyCot™, Fisher & Paykel Healthcare, Auckland, New Zealand) pre-warmed manually regulated overhead heater. Standardized room temperature in the delivery suite was 23–25 °C. If the neonate was below 28 weeks of gestation, it was put in a polyethylene bag. More mature preterm neonates were dried and had their bodies covered with warm towels. All neonates received elastic cotton caps to reduce heat loss from the head and were kept under the overhead heater for the measurement period of 15 min. Postnatal stabilization was performed according to current resuscitation guidelines [12,13]. SpO_2_ and HR were measured routinely using pulse oximetry on the right wrist or palm (IntelliVue MP50, Philips, Amsterdam, The Netherlands). Blood pressure was measured non-invasively at least once during postnatal stabilization. Respiratory support (continuous positive airway pressure [CPAP] and/or positive pressure ventilation [PPV]) was provided by using a silicone face mask (LSR Silicon mask no. 0/0 or 0/1; Laerdal Medical, Norway) and the Neopuff Infant T-Piece Resuscitator (Perivent, Fisher& Paykel Healthcare, Auckland, New Zealand). Requirement of respiratory support (CPAP and/or PPV or intubation) was documented. Non-heated and non-humidified gases were used for initial respiratory support. In minute 15 after birth, the rectal body temperature was measured once using a standard temperature probe (IntelliVue MP50, Philips, Amsterdam, The Netherlands).

### 2.3. Near-Infrared Spectroscopy

For cerebral NIRS measurements, the sensor was attached to the neonates’ right forehead, and for peripheral NIRS measurements, the sensor was placed on the right forearm by a member of the research team. The sensors were fixed with a gauze bandage. crSO_2_/cTOI and prSO_2_ were measured continuously during the first 15 min after birth. crSO_2_/cTOI was measured either with the INVOS 5100C device (Somanetics Corporation, Troy, Michigan, USA) or the NIRO 200-NX tissue oxygenation monitor (Hamamatsu Photonics, Hamamatsu-city, Japan). prSO_2_ was measured with the INVOS 5100C device. The peripheral fractional oxygen extraction (pFTOE) was calculated as follows: pFTOE = [(SpO_2_ − prSO_2_)/SpO_2_].

### 2.4. Data Collection

Vital parameters such as crSO_2_/cTOI, prSO_2_, HR, and SpO_2_, were saved automatically every second in a polygraphic data management system (Alpha-trace digital MM, BEST Medical Systems, Austria). These data were extracted and saved in a Microsoft Excel 2015 (Microsoft Corporation, Redmond, Washington, USA) database together with documented non-invasive blood pressure, rectal body temperature, need for respiratory support (CPAP and/or PPV or intubation), as well as prenatal and demographic information. Diagnoses (intraventricular hemorrhage [IVH], periventricular leukomalacia [PVL], NEC, and bacterial infection/sepsis) were extracted from electronic medical records (openMEDOCS, Version 6.5, SAP Business Client, Baden-Wuerttemberg, Germany).

### 2.5. Group Stratification

In a first step, the neonates were stratified into term neonates (≥37^+0^ weeks of gestation) and preterm neonates (<37^+0^ weeks of gestation). In a second step, sub-groups for preterm neonates were generated: (i) extremely preterm infants (EPI) born ≤28^+0^ weeks of gestation, (ii) very preterm infants (VPI) born between >28^+0^ and <32^+0^ weeks of gestation, and (iii) late preterm infants (LPI) born between ≥32^+0^ and <37^+0^ weeks of gestation. According to the WHO, body temperature stages were defined as follows: hyperthermia (>37.5 °C), normothermia (36.5–37.5 °C), mild hypothermia (36.0–36.4 °C), moderate hypothermia (32.0–35.9 °C), and severe hypothermia (<32.0 °C) [1].

### 2.6. Statistics

To detect and eliminate artifacts from the raw data, the following quality criteria for NIRS measurements were used. For physiological reasons, the SpO_2_, which is a measure of arterial hemoglobin oxygen saturation, should not be equal to or below the regional tissue saturation, which is a measure of hemoglobin oxygen saturation in the tissue including the arterial, capillary, and venous compartments [14]. If this was observed within the raw data, both arterial and regional oxygen saturation values were excluded from the analyses for abovementioned physiological reasons [14]. Data are given as mean ± standard deviation or as median (interquartile range) in dependency of distribution. Data were analyzed using IBM SPSS Statistics 24 Software (PSS Inc., Reston, VA, USA). Data distribution was tested using the Kolmogorov-Smirnov test. Mean values of crSO_2_/cTOI, prSO_2_, HR, and SpO_2_, were calculated for each minute after birth. For the statistics, we used crSO_2_/cTOI, prSO_2_, HR, and SpO_2_, values from minute 15 after birth. To investigate potential differences between groups, non-parametric tests (Mann-Whitney-U test and Kruskal Wallis test) were performed, and for the comparison of categorical variables, the χ^2^ test was used. For correlation analyses, we used Spearman’s correlation. A *p*-value of <0.05 was considered statistically significant.

## 3. Results

### 3.1. Characteristics

In a total of 587 neonates, continuous NIRS measurements and a single body temperature measurement within 15 min after birth were performed during the observation period. One neonate had to be excluded due to implausible data. Thus, 586 neonates (99.8%) were included in the present study. All 586 neonates received body temperature measurements, and in 492 neonates (84%), regional oxygen saturation values were available in minute 15 after birth. Four-hundred seventeen neonates (71.2%) were born at term and 169 neonates (28.8%) were born preterm. Demographic data are presented in detail in Table 1. Three-hundred two neonates (51.5%) were male, 50 neonates (8.5%) were twins, and six (1%) were triplets. One-hundred ninety-four neonates (33.1%) required respiratory support, thereof 188 neonates (32.1%) received CPAP and/or PPV, and 13 neonates (2.2%) were intubated during the observation period. Four of the included neonates (0.7%) needed cardiopulmonary resuscitation. 

### 3.2. Oxygenation

There was no association between body temperature and crSO_2_/cTOI and SpO_2_, a weak positive correlation between body temperature and HR in the whole cohort as well as a weak negative correlation between body temperature and prSO_2_ only in preterm infants. Correlation coefficients and *p*-values of body temperature and crSO_2_/cTOI, prSO_2_, pFTOE, HR, and SpO_2_ are demonstrated in detail in Table 2, Figure 1 and Figure 2. In neonates with hypothermia, we did not find any significant correlation between body temperature and crSO_2_/cTOI, prSO_2_, HR, and SpO_2_, (crSO_2_/cTOI: ρ = −0.02, *p* = 0.867; prSO_2_: ρ = −0.05, *p* = 0.672; HR: ρ = −0.177, *p* = 0.127; SpO_2_: ρ = 0.199, *p* = 0.069), respectively. In hyperthermic neonates, the body temperature did also not correlate significantly with crSO_2_/cTOI, prSO_2_, HR, and SpO_2_ (crSO_2_/cTOI: ρ = 0.111, *p* = 0.642; prSO_2_: ρ = 0.232, *p* = 0.658, HR: ρ = 0.300, *p* = 0.186; SpO_2_: ρ = −0.335, *p* = 0.118).

### 3.3. Body Temperature

The body temperature was statistically lower in preterm neonates compared to term neonates (median [IQR] 36.7 [36.4–37.0] °C vs. 36.8 [36.6–37.0] °C, *p* = 0.001). EPI had the highest body temperature (median [IQR] 37.0 [36.9–37.4] °C) and VPI (median [IQR] 36.7 [36.3–37.0] °C, *p* = 0.039) and LPI (median [IQR] 36.7 [36.4–37.0] °C, *p* = 0.017) had the lowest mean body temperature compared to the other groups 15 min after birth (Figure 3). There were significant differences in body temperature between LPI and term born infants (median [IQR] 36.7 [36.4–37.0] °C vs. 36.8 [36.6–37.0] °C, *p* = 0.004). There were no differences between LPI and VPI (*p* = 1.000), between VPI and term born infants (*p* = 0.279), as well as between EPI and term born infants (*p* = 0.293). In general, the prevalence of hypothermia was significantly higher in preterm neonates compared to term neonates (29.5% vs. 12.0%, *p* < 0.001). The prevalence of normothermic, hypothermic, and hyperthermic neonates within the groups and sub-groups is presented in detail in Table 3. None of the included neonates suffered from severe hypothermia. Four infants (0.7%) had a body temperature of >38.0 °C.

### 3.4. Respiratory Support

There was no difference in body temperature of the neonates who received respiratory support compared to those without need for respiratory support (mean 36.8 ± 0.4 °C vs. 36.8 ± 0.4 °C; *p* = 0.067). Among the term neonates, seventy-three (17.5%) needed respiratory support (CPAP and/or PPV). We found no significant difference in the body temperature of term neonates when comparing those with and those without the need for respiratory support (mean 36.8 ± 0.3 °C vs. 36.9 ± 0.4 °C, *p* = 0.544). One-hundred twenty-one preterm neonates (71.6%) required respiratory support, with no significant difference in body temperature between those who received respiratory support and those who did not (mean 36.7 ± 0.5 °C vs. 36.7 ± 0.4 °C, *p* = 0.764). 

### 3.5. Short-Term Outcome

Of the 586 included neonates, 26 (4.4%) suffered from proven early or late bacterial infection/sepsis and one neonate (0.2%) suffered from NEC. Thirteen neonates (2.2%) were diagnosed with IVH grade one and two neonates (0.3%) with IVH grade three. Five neonates (0.9%) developed PVL. Three patients (0.5%) died during the neonatal period. There was no difference in body temperature between neonates with one of the abovementioned adverse outcomes neither when testing for the combined endpoint (*p* = 0.238) nor for individual testing.

## 4. Discussion

In the present study, we described the association between body temperature and regional tissue oxygenation, HR, and SpO_2_ of term and preterm neonates born by Caesarean Section 15 min after birth. The main findings can be summarized as follows: (i) no association between body temperature and cerebral tissue oxygenation in term and preterm infants, (ii) no association between body temperature and peripheral tissue oxygenation in term infants, (iii) a weak negative correlation between body temperature and peripheral tissue oxygenation in preterm infants, (iv) a weak positive correlation between body temperature and HR in term and preterm infants, and (v) no association between body temperature and SpO_2_ in term and preterm infants. 

The absent association between body temperature and cerebral tissue oxygenation may be explained by the neonatal brain’s autoregulation ability and by the only mild deviation of body temperature in our cohort. This assumption is supported by a recently published porcine model study, investigating the changes of cerebral autoregulation during induction of deep hypothermia [15]. A rapid induction of deep hypothermia was performed to simulate the clinical scenario of accidental hypothermia. It was demonstrated that the analyzed autoregulation indices (pressure reactivity index, oxygen reactivity index, brain tissue oxygen tension, and the cerebral oximetry index) reflected normal cerebral autoregulation and did not change until a brain temperature of 34 °C. Cerebral tissue oxygen saturation increased not before a brain temperature of 29 °C [15]. However, cerebrovascular autoregulation varies between term and preterm neonates. Pressure reactivity and autoregulation to systolic and MABP are not observed until 26 to 28 weeks of gestation due to the on-going vascular development and anatomical features of the premature cerebral vasculature [16]. The very small number of EPI in our cohort may explain the absent association between body temperature and cerebral tissue oxygen saturation. 

We found a weak, but significantly negative correlation between peripheral tissue oxygen saturation and central body temperature only in preterm neonates. However, as cardiac function, blood gases (e.g., carbon dioxide), hemoglobin levels, and especially peripheral temperature all influence peripheral blood flow, tissue oxygenation, and oxygen extraction, interpretation of this finding is challenging [17]. The weak negative correlation may be explained by the fact that hyperthermia causes a higher metabolic rate, resulting in higher tissue oxygen extraction and a consequent decrease in peripheral tissue oxygen saturation. Furthermore, microvascular perfusion in preterm infants may be more affected by temperature variations than peripheral perfusion in term neonates, due to their higher body surface to weight ratio, higher heat loss by evaporation due to skin immaturity, deficiency of subcutaneous adipose tissue, and limited ability to regulate cutaneous blood flow [18,19].

Physiologically, the HR of neonates should decrease when suffering from hypothermia, at least initially [20]. Thereafter, norepinephrine gets released and the HR increases [20]. This matches our finding of a significantly positive correlation between body temperature and HR for the whole cohort as well as in preterm and term neonates individually. This was also observed in other studies [2,21], whereby Davies et al. reported the increase in HR per added degree of temperature is approximately 10 bpm based on observations in 31851 children, attending the pediatric emergency department [21]. On the other hand, we did not find a correlation between body temperature and SpO_2_, which was independent from gestational age. Literature about the association between body temperature and SpO_2_ in neonates is scarce. Mitra et al. demonstrated that SpO_2_ dropped briefly at the start of induced therapeutic hypothermia in neonates, but it was mostly above 95% [22]. When the body temperature was raised to 37 °C, SpO_2_ remained stable [22], suggesting no clinically relevant association between body temperature and SpO_2_. This was confirmed in a recent study by Wu et al., where SpO_2_ remained stable during the rewarming phase after induced hypothermia in neonates with hypoxic-ischemic encephalopathy [23].

Regarding our secondary aim, we observed statistically lower body temperature in preterm neonates; however, the difference was small (mean of 0.1 °C) and we expect that its clinical relevance is negligible. Nevertheless, preterm neonates suffered from hypothermia significantly more often compared to term neonates. There are only limited data about hypothermia in term neonates. We reported hypothermia only in 12% of term born infants, whereas Takayama et al. observed that 17% of the included healthy term born infants suffered from hypothermia 34 min after birth [6]. In contrast to term born infants, several studies have investigated body temperature in large cohorts of preterm infants. The incidence of hypothermia in preterm neonates ranged between 38–53% [9,24,25], and in EPI the incidence of hypothermia below 35.0 °C was found in a worrying 9.6% [25]. In contrast, Lyu et al. reported that only 12% of the investigated preterm infants had an admission temperature of below 36.0 °C, and only 2% had an admission temperature lower than 35.0 °C [3]. This suggests that temperature management varies between institutions and may have a profound impact on postnatal body temperature.

We observed a distinctly lower rate of hypothermia in our cohort compared to the results of the abovementioned studies, as a body temperature of below 36.0 °C was only found in 4.1% of preterm infants. The highest rate of hypothermia occurred in VPI (7.9%), but even this rate is considerably lower than in other studies. Interestingly, none of the EPI had a body temperature of below 36.0 °C. We speculate that attention to body temperature management was paid mostly to EPI and to a lesser extent to more mature preterm neonates. Only Lyu et al. published an incidence of hypothermia in preterm infants similar to our results, but it has to be acknowledged that their hypothermia thresholds were lower than ours [3]. The number of included infants, the earlier body temperature measurement, and the fact that this is a mono-centric study may explain the differences between our results and those of other studies. In the present study, all included infants were born by Caesarean section. The birth mode may have an impact on vascular reactivity, peripheral oxygenation, and thermal homeostatic mechanism. Additionally, vaginally delivered newborns often benefit from postnatal skin-to-skin contact, whereas infants born by Caesarean section are regularly placed on a resuscitation table after birth, resulting in a different body temperature management. Additionally, in all included infants, early cord clamping (<30 s) was performed. Timing of umbilical cord clamping has an impact on neonatal hemodynamics and may affect both body temperature and cerebral tissue oxygenation. Whereas studies reported no differences in cerebral tissue oxygenation six to 12 h after birth when comparing immediate and delayed cord clamping [26], Pichler et al. demonstrated that delayed cord clamping caused lower initial cerebral tissue oxygen saturation in spontaneously breathing preterm neonates compared to preterm neonates without immediate cord clamping [27,28]. Furthermore, we observed a high rate of normothermic infants and, therefore, temperature differences may have been more pronounced in other studies.

Several studies demonstrated that the number of preterm infants suffering from hypothermia could be decreased (without increasing hyperthermia rates) by using a practice plan including consistent head and torso wrapping, warmed blankets, a transwarmer mattress, and maintaining a consistent operating room temperature (between 21 °C and 23 °C) [29,30]. In consideration of our observations, we assume that our standardized postnatal temperature management concept seems to be beneficial for body temperature preservation in preterm infants. 

Besides hypothermia, also iatrogenic hyperthermia is a complication with detrimental neonatal outcomes [3]. Overheating can be caused by the use of plastic wraps, radiant warmers, incubators, or excessive environment heating [10,31]. Studies described hyperthermia incidences in preterm infants at admission between 1–2% [3,7,25]. In comparison to these studies, the prevalence of hyperthermia was higher in our study, yet the difference was small. According to the WHO, we defined hyperthermia as a body temperature of above 37.5 °C, which is in contrast to the abovementioned studies, defining hyperthermia mostly ≥38.0 °C. This different definition may explain the higher rate of hyperthermia in our cohort. Nevertheless, it has to be considered that two of the included ten EPI suffered from hyperthermia. This may have been caused by the use of polyethylene bags [32,33]. However, Lenclen et al. demonstrated that a higher admission body temperature may be achieved in preterm neonates without increasing the risk for hyperthermia by using polyethylene bags [34]. Additionally, maternal fever and/or infections such as chorioamnionitis may have contributed to the rate of neonatal hyperthermia, but these data were not available in our database. Nonetheless, it should be kept in mind that excluding neonates from mothers with signs of infection would have increased the incidence of neonates suffering from hypothermia.

The use of heated humidified gases for respiratory support to avoid hypothermia in newborn infants is an important concept. A recent meta-analysis investigating heated humidified gases for respiratory support in preterm neonates during resuscitation could only include two studies [35]. Still, the authors concluded that heating and humidification of inspired gases immediately after birth improves admission temperature in preterm infants [35]. In the present study, we did not find any differences between newborn infants without respiratory support and those who received respiratory support using non-heated unconditioned gases. This raises the question whether the use of heated humidified gases offers a substantial advantage during immediate postnatal stabilization and resuscitation in a cohort of newborn infants in whom the hypothermia rate was generally low. 

### Limitations and Strengths

There are some limitations associated with this study. Firstly, this is a single center study. Secondly, we performed only one body temperature measurement in each patient and therefore have no data about the course of body temperature. Thirdly, we only recorded the need for respiratory support, but data about the fraction of inspired oxygen were not available in our data base. All infants were born by Caesarean section, which should be considered when interpreting the results. The small number of both extremely premature neonates and of adverse clinical outcomes may have contributed towards some expected or apparent associations having reached no statistical significance. Finally, a bias might be that all of the included neonates participated in clinical studies and as a result more attention may have been paid to body temperature management and preservation.

This is the first study investigating the relationship between cerebral and peripheral tissue oxygen saturation and central body temperature in neonates 15 min after birth, thus adding important novel data to the field. Further strengths of the study are the direct comparison between term and preterm neonates, the sub-group analysis of the preterm neonates, and the consistent use of rectally measured, i.e., central, body temperature. Several studies have shown that axillary body temperature measurements tend to be lower than those gained rectally [5,36,37] and that the rectal body temperature is a more accurate measure of body core temperature [1]. 

## 5. Conclusions

Body temperature neither correlated with crSO_2_/cTOI nor with SpO_2_ and this finding was independent from gestational age. We found a weak negative correlation between prSO_2_ and body temperature in preterm neonates and a weak positive correlation between HR and body temperature in the whole cohort, as expected. Body temperature during immediate postnatal transition was statistically lower in preterm neonates compared to term neonates, but the difference was small and probably clinically irrelevant. Nevertheless, hypothermia was significantly more often observed in preterm neonates than in to term neonates. Preterm infants >28^+0^ and <32^+0^ weeks of gestation had the highest prevalence of hypothermia, whereas hyperthermia was most frequent in preterm infants born below 28^+0^ weeks of gestation. Therefore, attention to stringent body temperature management should be paid not only to extremely preterm neonates but also to very and late preterm infants.

## Figures and Tables

**Figure 1 children-07-00205-f001:**
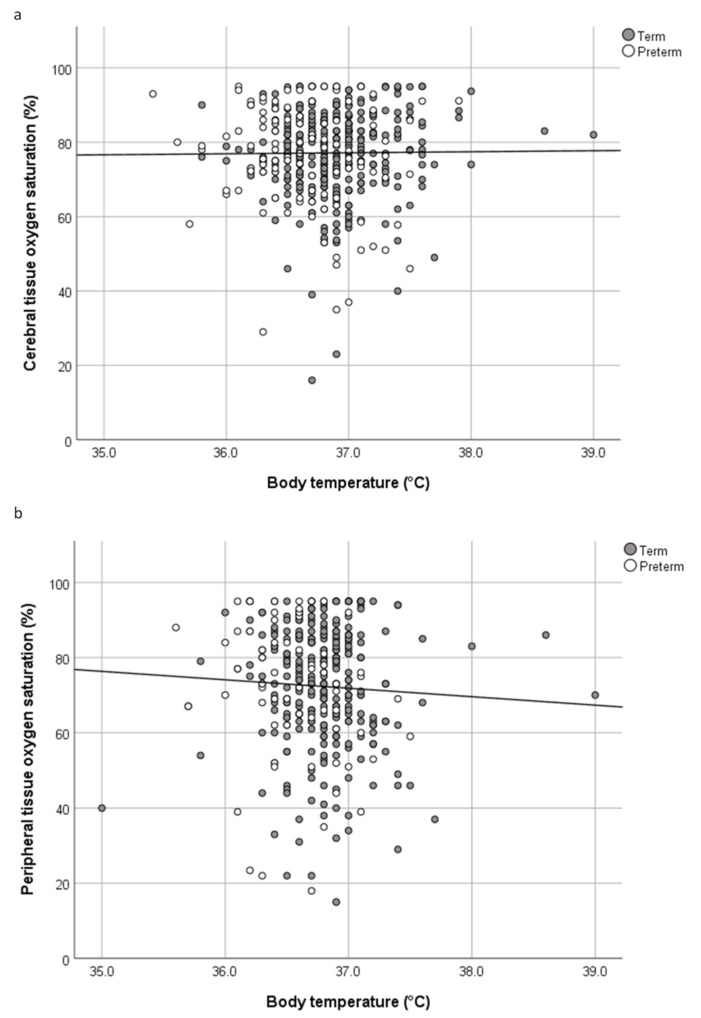
(**a**) Scatter plot of cerebral tissue oxygen saturation (%) and body temperature (°C) in term and preterm neonates. (**b**) Scatter plot of peripheral tissue oxygen saturation (%) and body temperature (°C) in term and preterm neonates.

**Figure 2 children-07-00205-f002:**
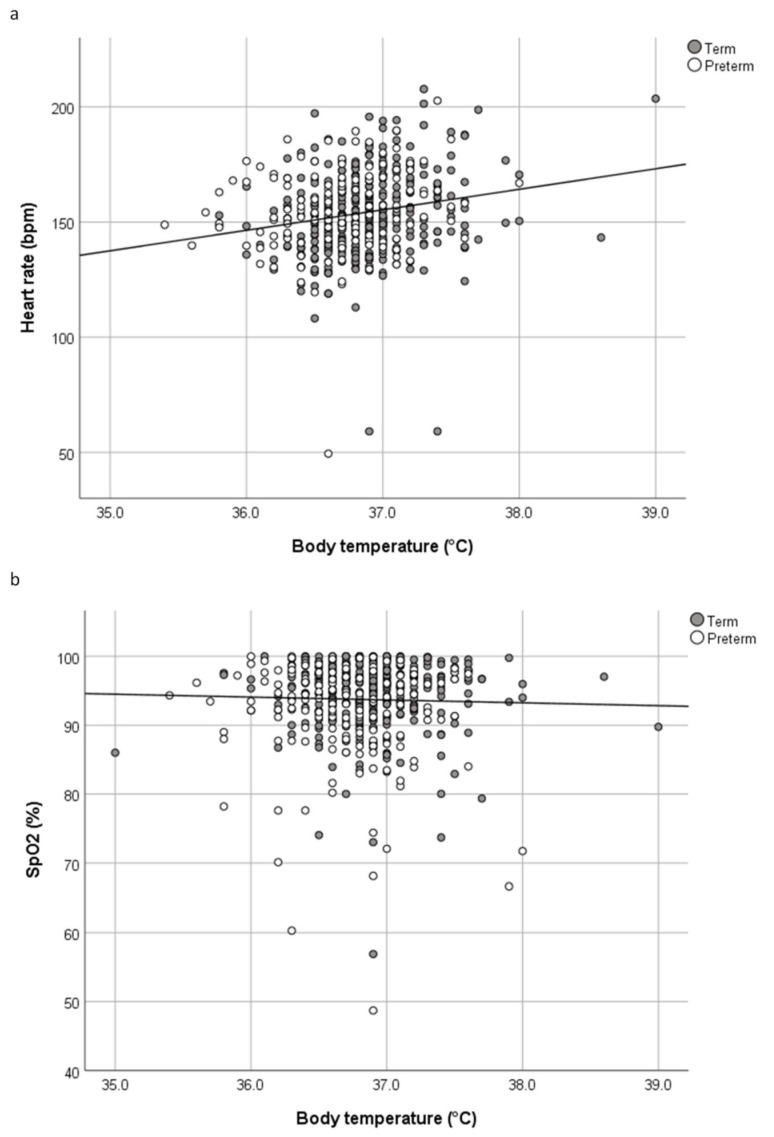
(**a**) Scatter plot of heart rate in beats per minute (bpm) and body temperature (°C) in term and preterm neonates. (**b**) Scatter plot of arterial oxygen saturation (SpO_2_)% and body temperature (°C) in term and preterm neonates.

**Figure 3 children-07-00205-f003:**
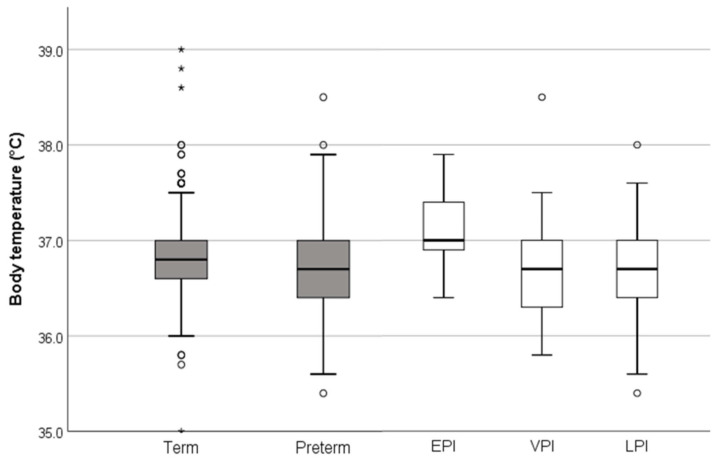
Boxplot (box: median, 1st and 3rd quartile; whisker: lower and upper extreme within 1.5 times the inter-quartile range from the upper or lower quartile; O: outliner >1.5 to <3 times the interquartile range from the upper or lower quartile, *: extreme outliner ≥3 times the interquartile range from the upper or lower quartile; highest and lowest extreme outliner demonstrate minimum and maximum range) of body temperature (°C) according to gestational age: term (≥37^+0^ weeks of gestation), preterm (<37^+0^ weeks of gestation) and subgroups EPI (extremely preterm infants ≤28^+0^ weeks of gestation), VPI (very preterm infants >28^+0^ and <32^+0^ weeks of gestation), and LPI (late preterm infants ≥32^+0^ and <37^+0^ weeks of gestation).

**Table 1 children-07-00205-t001:** Demographic data of the whole cohort, preterm and term born neonates, as well as of the subgroups EPI (extremely preterm infants ≤28^+0^ weeks of gestation), VPI (very preterm infants >28^+0^ and <32^+0^ weeks of gestation), and LPI (late preterm infants ≥32^+0^ and <37^+0^ weeks of gestation). Data are expressed as mean ± standard deviation or median (interquartile range). Gestational age (GA), birth weight (BW), umbilical arterial pH (pHa), mean arterial blood pressure (MABP), arterial oxygen saturation (SpO_2_), heart rate in beats per minute (bpm), body temperature in degrees Celsius (°C), cerebral tissue oxygen saturation (crSO_2_/cTOI), peripheral tissue oxygen saturation (prSO_2_), and peripheral fractional tissue oxygen extraction (pFTOE) in minute 15.

	Total (*n* = 586)	Term (*n* = 417)	Preterm (*n* = 169)	*p*-Value	EPI (*n* = 10)	VPI (*n* = 38)	LPI (*n* = 121)
**GA (weeks)**	39 (36–39)	39 (38–39)	34 (32–35)	<0.001	26 (24–27)	31 (30–31)	34 (33–36)
**Apgar 1**	9 (8–9)	9 (9–9)	8 (8–9)	<0.001	8 (8–9)	8 (7–8)	8 (8–9)
**Apgar 5**	10 (9–10)	10 (10–10)	9 (8–10)	<0.001	8 (6–8)	9 (8–9)	9 (9–10)
**Apgar 10**	10 (10–10)	9 (9–10)	10 (10–10)	<0.001	9 (9–10)	9 (9–10)	10 (9–10)
**BW (grams)**	3088 (2380–3460)	3290 (2996–3590)	1830 (1438–2260)	<0.001	700 (626–906)	1330 (1054–1564)	2040 (1756–2400)
**pHa**	7.30 (7.28–7.32)	7.30 (7–28-7.32)	7.31 (7.28–7.34)	0.004	7.33 (7.31–7.39)	7.32 (7.29–7.34)	7.30 (7.27–7.33)
**MABP (mmHg)**	45 ± 10	47 ± 9	40 ± 9	<0.001	31 ± 7	38 ± 8	41 ± 9
**SpO_2_ (%)**	95.1 (91.7–97.7)	95.7 (92.7–97.9)	93.5 (88.5–96.8)	<0.001	92.6 (72.1–96.1)	90.4 (86.9–93.8)	94.5 (91.1–97.3)
**Heart rate (bpm)**	154 ± 18	153 ± 18	156 ± 18	0.015	159 ± 17	161 ± 16	154 ± 18
**Body temperature (°C)**	36.8 (36.6–37.0)	36.8 (36.6–37.0)	36.7 (36.4–37.0)	0.001	37.0 (36.9–37.4)	36.7(36.3–37.0)	36.7 (36.4–37.0)
**crSO_2_/cTOI (n)**	458	342	116		5	22	89
**crSO_2_/cTOI (%)**	78.9 (71.0–86.0)	78.4 (71.0–85.0)	79.0 (68.7–87.5)	0.853	79.6 (75.6–91.1)	71.7 (62.0–79.0)	80.0 (72.0–89.0)
**prSO_2_ (n)**	355	289	66		1	9	56
**prSO_2_ (%)**	74.0 (63.0–85.0)	74.0 (63.0–85.0)	73.0 (62.0–87.0)	0.728	69.0 (69.0–69.0)	65.0 (51.0–75.0)	76 (63–87.5)
**pFTOE**	0.23 (0.12–0.34)	0.22 (0.12–0.34)	0.23 (0.11–0.30)	0.596	0.28 (0.28–0.28)	0.24 (0.16–0.40)	0.23 (0.09–0.29)

**Table 2 children-07-00205-t002:** Correlation analyses of body temperature in degrees Celsius (°C) and cerebral tissue oxygen saturation (crSO_2_/cTOI), peripheral tissue oxygen saturation (prSO_2_), peripheral fractional tissue oxygen extraction (pFTOE), heart rate in beats per minute (bpm), and arterial oxygen saturation (SpO_2_).

	Total Cohort (*n* = 586)		Term (*n* = 417)		Preterm (*n* = 169)	
	Body Temperature (°C)	*p*-Value	Body Temperature (°C)	*p*-Value	Body Temperature (°C)	*p*-Value
crSO_2_/cTOI (%)	ρ = 0.018	0.701	ρ = 0.077	0.155	ρ = −0.102	0.275
prSO_2_ (%)	ρ = −0.092	0.084	ρ = −0.044	0.461	ρ = −0.285	0.020
pFTOE	ρ = 0.042	0.443	ρ = 0.011	0.852	ρ = 0.165	0.195
Heart rate (bpm)	ρ = 0.210	<0.001	ρ = 0.236	<0.001	ρ = 0.222	<0.006
SpO_2_ (%)	ρ = −0.015	0.741	ρ = −0.013	0.805	ρ = −0.125	0.111

**Table 3 children-07-00205-t003:** Prevalence of normothermia (36.5–37.5 °C), mild hypothermia (36.0–36.4 °C), moderate hypothermia (32.0–35.9 °C), and hyperthermia (>37.5 °C) in the study sample (term [≥37^+0^ weeks of gestation], preterm [<37^+0^ weeks of gestation], and subgroups EPI [extremely preterm infants ≤28^+0^ weeks of gestation], VPI [very preterm infants >28^+0^ and <32^+0^ weeks of gestation], and LPI [late preterm infants ≥32^+0^ and <37^+0^ weeks of gestation]). Data are expressed in *n* (%).

	Total (*n* = 586)	Term (*n* = 417)	Preterm (*n* = 169)	EPI (*n* = 10)	VPI (*n* = 38)	LPI (*n* = 121)
**Normothermia**	461 (78.7)	347 (83.2)	114 (67.5)	7 (70.0)	23 (60.5)	84 (69.4)
**Mild hypothermia**	89 (15.2)	46 (11.0)	43 (25.4)	1 (10.0)	11 (29.0)	31 (25.6)
**Moderate hypothermia**	11 (1.9)	4 (1.0)	7 (4.1)	0 (0.0)	3 (7.9)	4 (3.3)
**Hyperthermia**	25 (4.3)	20 (4.8)	5 (3.0)	2 (20.0)	1 (2.6)	2 (1.7)

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
