# Peer review of "Association between Regional Tissue Oxygenation and Body Temperature in Term and Preterm Infants Born by Caesarean Section"

_children, 2020, doi:10.3390/children7110205_

Round 1

Reviewer 1 Report

Dear Authors,

The subject of manuscript “Body temperature and oxygenation in term and preterm infants during fetal-to-neonatal transition” is important to neonatology. Hypothermia at NICU admission is an important quality mark of neonatal care at birth and it is independently associated with morbidity and mortality. The study of the association of body temperature after birth with peripheral and brain oxygenation is original and may bring new information to the understanding of the complex interplay of the different systems to maintain homeostasis during a period that homeostasis means change, that is, during the transitional phase soon after birth.  

In this context, the study is extremely interesting, but it is not focused on its main contribution.  This is a major issue: the novelty of the study is the analysis of the association between brain/tissue oxygenation and hypothermia 15 minutes after birth. However, although the title offers to the reader this flavor, the primary objective, the main outcome, the results, and the discussion deal mainly with something not new and already known – the frequency of hypothermia according to gestational age. As a “neutral reader”, I had the impression that, as a consequence of finding no association between brain oxygenation and hypothermia at 15 minutes, the authors tried to highlight the finding that “preterm neonates had a significantly lower BT and suffered significantly more often from hypothermia during postnatal transition” (which is emphasized in the abstract, results, discussion and conclusion). Well, this is already known and analyzed by several studies (referred by the authors), which were designed specifically to study thermal homeostasis and thermal care soon after birth. This was not the case of the present study, which is a secondary analysis of 5 other studies made with different objectives and it is one of the reasons of the main weakness of the present analysis: the fact that the only available data on temperature was the body temperature at 15 minutes of life. Despite this huge weakness, the authors do have interesting and original data to at least make an analysis that generate hypothesis and new research. Therefore, I think that the manuscript should be completely revised to highlight the association (or lack of association) between brain and body oxygenation and body temperature at 15 minutes after birth. Everything else is secondary and all the excessive data on frequency of hypothermia on different gestational ages should be dealt as characteristics of the sample and maybe as adjusting variables for analysis, but not as a main outcome (the authors have 10 extremely preterm infant…).

Minor issues:

  • Line 33: In the sentence “However, the literature about normal body temperature is inconsistent and the normal range recommended by the WHO is under discussion nowadays [2](3).” – The references are 15-year old and the “nowadays” seems out of place.
  • Line 82: All patients were born by C-section: this may have had an impact on results, such as vascular reactivity, peripheral oxygenation, and thermal homeostatic mechanisms. This fact should be discussed and acknowledge as a study limitation.
  • Line 93: The authors report the blood pressure was measures for all patients during stabilization in DR. Is this important for this study? If so, I was intrigued how they managed to measure the BP during this period since they still did not have lines to measure invasive BP and the oscillometric method does not seem very easy to apply during the first minutes after birth.
  • Results: text, tables and figures are repetitive.
  • Line 153: The authors report that “188 neonates (32.1%) received CPAP and/or PPV” –it seems completely different for this study the patients that needed help to establish adequate ventilation after birth (NB that needed PPV) from those that needed help to establish FRC or to maintain oxygenation (NB that needed CPAP). These different patients may have diverse modes of adaptation in terms of brain oxygenation and thermal homeostasis.
  • Table 1 (demographic data) – the authors do not comment on the losses of preterm patients measuring brain oxygenation (50% of the EPI and 40% of the VPI) and tissue oxygenation (90% of the EPI and 75% of VPI) – what is the internal validity of the results obtained for this group with so many losses? I would analyze the preterm group as a whole, without subdividing them (there is no power for these divisions).

In conclusion, the data is very interesting, but the authors should thoroughly revise the whole text in order to address the original data that they collected regarding the association of body temperature soon after birth and brain and tissue oxygenation.

Author Response

Dear Authors,

The subject of manuscript “Body temperature and oxygenation in term and preterm infants during fetal-to-neonatal transition” is important to neonatology. Hypothermia at NICU admission is an important quality mark of neonatal care at birth and it is independently associated with morbidity and mortality. The study of the association of body temperature after birth with peripheral and brain oxygenation is original and may bring new information to the understanding of the complex interplay of the different systems to maintain homeostasis during a period that homeostasis means change, that is, during the transitional phase soon after birth.  

In this context, the study is extremely interesting, but it is not focused on its main contribution.  This is a major issue: the novelty of the study is the analysis of the association between brain/tissue oxygenation and hypothermia 15 minutes after birth. However, although the title offers to the reader this flavor, the primary objective, the main outcome, the results, and the discussion deal mainly with something not new and already known – the frequency of hypothermia according to gestational age. As a “neutral reader”, I had the impression that, as a consequence of finding no association between brain oxygenation and hypothermia at 15 minutes, the authors tried to highlight the finding that “preterm neonates had a significantly lower BT and suffered significantly more often from hypothermia during postnatal transition” (which is emphasized in the abstract, results, discussion and conclusion). Well, this is already known and analyzed by several studies (referred by the authors), which were designed specifically to study thermal homeostasis and thermal care soon after birth. This was not the case of the present study, which is a secondary analysis of 5 other studies made with different objectives and it is one of the reasons of the main weakness of the present analysis: the fact that the only available data on temperature was the body temperature at 15 minutes of life. Despite this huge weakness, the authors do have interesting and original data to at least make an analysis that generate hypothesis and new research. Therefore, I think that the manuscript should be completely revised to highlight the association (or lack of association) between brain and body oxygenation and body temperature at 15 minutes after birth. Everything else is secondary and all the excessive data on frequency of hypothermia on different gestational ages should be dealt as characteristics of the sample and maybe as adjusting variables for analysis, but not as a main outcome (the authors have 10 extremely preterm infant…).

Minor issues:

  • Line 33: In the sentence “However, the literature about normal body temperature is inconsistent and the normal range recommended by the WHO is under discussion nowadays [2](3).” – The references are 15-year old and the “nowadays” seems out of place.

The reviewer is right and we have adjusted the sentence.

  • Line 82: All patients were born by C-section: this may have had an impact on results, such as vascular reactivity, peripheral oxygenation, and thermal homeostatic mechanisms. This fact should be discussed and acknowledge as a study limitation.

We agree and have added and acknowledged it in the discussion (Line 301) and limitations (Line 349).

  • Line 93: The authors report the blood pressure was measures for all patients during stabilization in DR. Is this important for this study? If so, I was intrigued how they managed to measure the BP during this period since they still did not have lines to measure invasive BP and the oscillometric method does not seem very easy to apply during the first minutes after birth.

Our nursing staff is well trained in performing oscillometric BP measurements within the first minutes after birth, which is standard in our delivery room for sick infants. We reported on blood pressure in the manuscript for the sake of completeness and to demonstrate the circulatory stability of our study population.

  • Results: text, tables and figures are repetitive.

We have tried to eliminate repetitive information from the manuscript, wherever evident:

  • 3.1. Characteristics: No repetitions in tables or figures
  • 3.2. Oxygenation: We have described the results of our primary outcome in one short sentence (details of correlations are demonstrated in Table 2). The correlation coefficients of neonates with hypothermia and hyperthermia are only described in that paragraph and not anywhere else.
  • 3.3. Body Temperature: In this paragraph we compared the different gestational age groups with each other and described the p-values of group comparison. Presenting the p-values in Table 1 would have been too confusing, so we chose to give them in this paragraph.
  • 3.4. Respiratory support: No repetitions in tables or figures
  • 3.5. Short-term outcome: No repetitions in tables or figures

  • Line 153: The authors report that “188 neonates (32.1%) received CPAP and/or PPV” –it seems completely different for this study the patients that needed help to establish adequate ventilation after birth (NB that needed PPV) from those that needed help to establish FRC or to maintain oxygenation (NB that needed CPAP). These different patients may have diverse modes of adaptation in terms of brain oxygenation and thermal homeostasis.

The reviewer is completely right. We have already addressed this aspect in the limitations (Line 347: “Thirdly, we did only record the need for respiratory support, but data about the fraction of inspired oxygen were not available in our database.”) Unfortunately, due to the lack of additional data regarding respiratory support, we can only speculate about this relevant factor.

  • Table 1 (demographic data) – the authors do not comment on the losses of preterm patients measuring brain oxygenation (50% of the EPI and 40% of the VPI) and tissue oxygenation (90% of the EPI and 75% of VPI) – what is the internal validity of the results obtained for this group with so many losses? I would analyze the preterm group as a whole, without subdividing them (there is no power for these divisions).

We included neonates who received continuous pulse oximetry and NIRS monitoring during the first 15 minutes after birth as well as a single body temperature measurement in minute 15 after birth. For data analyses we used tissue oxygen saturation values from minute 15; however, as this study was retrospective, regional oxygen saturation values were not available in minute 15 for all infants. We reported on that in Line 139. In the preterm subgroups we only compared body temperature with each other. For correlation analysis regarding body temperature and regional tissue oxygenation, HR, and SpO2, we analyzed only the groups term and preterm (and no preterm subgroups), as suggested by the reviewer (Table 2, Figures 1 and 2).

In conclusion, the data is very interesting, but the authors should thoroughly revise the whole text in order to address the original data that they collected regarding the association of body temperature soon after birth and brain and tissue oxygenation.

These concerns are in concordance with those of the other reviewers, who questioned the clarity of goals and study objective and suggested to highlight the association (or lack of an association) between brain and muscle tissue oxygenation and body temperature. Accordingly, we decided to focus on the associations between body temperature and NIRS parameters as our primary outcome measure, and to report on already well known differences in body temperature between term and preterm infants as secondary outcome. We have thoroughly revised the whole manuscript and hope that the manuscript has improved in clarity of goals and objective.

Thank you for the valuable and supportive review!

Reviewer 2 Report

Thank you for the opportunity to review “Body temperature and oxygenation in term and preterm infants during fetal-to-neonatal transition”. The manuscript is easy and pleasant to read, having interesting contributions to the field.

Overall, the appreciation is rather positive. There are some conceptual and operational, methodological issues that deserve further clarification and, eventually, corrections.

The study is a post-hoc, secondary analysis that uses data from five previous studies. Those data were collected for different proposes from the aims of the study the manuscript presents. Therefore, the manuscript should explicitly acknowledge its post-hoc, secondary analysis nature; as a collateral effect, it has to substitute the formula “during the study period” for a more adequate expression, such as “during the observation period”.

The study collected data from samples of neonates born by caesarean section only. This must be explicitly considered in the Discussion (maybe also in the Title) and acknowledged when discussing the comparability and the limits to the generalization of the obtained results.

The manuscript states that only neonates born with “congenital malformations that could potentially affect cardiorespiratory or neurological function” were excluded. Therefore, it can be assumed that every other neonate born by caesarean section was included, in spite of the cause of the caesarean section and the maternal condition. This being truth, several important confounding conditions and eventual bias are not considered, as emergency caesarean section, acute foetal distress, chorioamnionitis and other causes of maternal fever, as well as relatively frequent maternal conditions, as those affecting the thyroid. In fact, there is a relatively large number of neonates in the study that had body temperature >38ºC. An effort should be made to control or to take into account these confounders and biases and it must be explicit in the manuscript.

The description of the strategy for statistical analysis is clear but not its factual implementation. Body temperature, the core variable in the study, is described in Table 1 and in Figure 1 using median and interquartile range, which is very informative and suggests the authors followed the assumptions of non-Normality planned on the strategy for statistical analysis. When comparing body temperature between groups and when assessing its association with other exposure or effect variables, it is always presented as mean and standard deviation, suggesting the comparisons were made using parametric tests, rather than the non-parametric tests presented on the strategy for statistical analysis. These inconsistencies may reflect either a bad choice of presenting the data from the analysis or a wrong statistical analysis. Either possibility requires corrections but if inadequate statistic tests were used, the analysis must be done again and every result must be corrected; their interpretation might change, as well as some of the conclusions presented in the manuscript.

When assessing the association of body temperature with some variables with more than two categories, only correlation coefficients are presented; in fact, it would be an insufficient analysis and either ANOVA or Kruskal-Wallis tests, as adequate, should be performed and presented.

The results of some of the correlation analysis are presented in the Results section as “positive (or negative) correlation”; the values of r advise to add “positive (or negative) weak correlation”. On the other hand, the differences in the values of body temperature are referred in the Discussion as being “significantly lower” and it is later clarified that the difference might not have any clinical significance; it would be wise to use the formula “statistically lower” instead of “significantly lower”, to avoid ambiguity.

The small number of both extremely premature neonates and of adverse clinical outcomes may have contributed for some expected or apparent associations having reached no statistical significance; it should be acknowledges in the Discussion.

From my point of view, Figure 2 does not add anything to Table 2.; Figure 1 requires better explanation of the content of the graphic and the graphics in Figure 3 and 4 might improve adding regression lines (ou curves).

Some minor comments and proposals of corrections are to be found in the annexed document but I would like to highlight that there is no evidence of a rationale to use "randomized controlled trials" to assess an observation of physiologic thresholds (line 40).

Author Response

Thank you for the opportunity to review “Body temperature and oxygenation in term and preterm infants during fetal-to-neonatal transition”. The manuscript is easy and pleasant to read, having interesting contributions to the field.

Overall, the appreciation is rather positive. There are some conceptual and operational, methodological issues that deserve further clarification and, eventually, corrections.

The study is a post-hoc, secondary analysis that uses data from five previous studies. Those data were collected for different proposes from the aims of the study the manuscript presents. Therefore, the manuscript should explicitly acknowledge its post-hoc, secondary analysis nature; as a collateral effect, it has to substitute the formula “during the study period” for a more adequate expression, such as “during the observation period”.

The reviewer is right. We have acknowledged that this study is a post-hoc secondary outcome parameter analysis (Line 60). We have substituted the formula “during the study period” to “during the observation period” through the whole manuscript.

The study collected data from samples of neonates born by caesarean section only. This must be explicitly considered in the Discussion (maybe also in the Title) and acknowledged when discussing the comparability and the limits to the generalization of the obtained results.

We have added and acknowledged this relevant factor in the limitations (Line 349) and the discussion (Line 301) as well as in the title.

The manuscript states that only neonates born with “congenital malformations that could potentially affect cardiorespiratory or neurological function” were excluded. Therefore, it can be assumed that every other neonate born by caesarean section was included, in spite of the cause of the caesarean section and the maternal condition. This being truth, several important confounding conditions and eventual bias are not considered, as emergency caesarean section, acute foetal distress, chorioamnionitis and other causes of maternal fever, as well as relatively frequent maternal conditions, as those affecting the thyroid. In fact, there is a relatively large number of neonates in the study that had body temperature >38ºC. An effort should be made to control or to take into account these confounders and biases and it must be explicit in the manuscript.

We fully agree with the reviewer that the indication why to perform Caesarean section may have impacted our results. However, the reasons why included neonates were born by Caesarean section were not summarized sufficiently in our database and, thus, we can only speculate about the impact of different maternal-fetal conditions. Still, 4.3% of the included infants were hyperthermic when using the stringent definition of  >37.5°C, and only four (0.7%) of all included infants had a body temperature of  >38.0°C. Hence, we assume that maternal fever/infection did not have a relevant impact on our results. Nonetheless, we have added the potential impact of maternal-fetal infections to the discussion about hyperthermia (Line 331).

The description of the strategy for statistical analysis is clear but not its factual implementation. Body temperature, the core variable in the study, is described in Table 1 and in Figure 1 using median and interquartile range, which is very informative and suggests the authors followed the assumptions of non-Normality planned on the strategy for statistical analysis. When comparing body temperature between groups and when assessing its association with other exposure or effect variables, it is always presented as mean and standard deviation, suggesting the comparisons were made using parametric tests, rather than the non-parametric tests presented on the strategy for statistical analysis. These inconsistencies may reflect either a bad choice of presenting the data from the analysis or a wrong statistical analysis. Either possibility requires corrections but if inadequate statistic tests were used, the analysis must be done again and every result must be corrected; their interpretation might change, as well as some of the conclusions presented in the manuscript.

The reviewer is right. We used the Kruskal-Wallis test for group comparisons as the data were not normally distributed, but the data were presented in mean (SD). We have changed the data presentation to median (IQR) in paragraph “3.3 Body temperature (Line 188-199)”, as suggested.

When assessing the association of body temperature with some variables with more than two categories, only correlation coefficients are presented; in fact, it would be an insufficient analysis and either ANOVA or Kruskal-Wallis tests, as adequate, should be performed and presented.

According to suggestions from a medical statistician, we assessed the association of body temperature only in two categories (term and preterm). For this, we used Spearman´s correlation analysis and presented the correlation coefficients and p-values. The associations between body temperature with some variables in hyperthermic infants and hypothermic infants were also performed separately with Spearman’s correlation. To compare body temperature of the different groups (term, preterm, EPI, VPI and LPI), we used the Kruskal-Wallis test, as the data were not normally distributed. Further statistical tests, including the mentioned ANOVA, were not planned initially.

The results of some of the correlation analysis are presented in the Results section as “positive (or negative) correlation”; the values of r advise to add “positive (or negative) weak correlation”. On the other hand, the differences in the values of body temperature are referred in the Discussion as being “significantly lower” and it is later clarified that the difference might not have any clinical significance; it would be wise to use the formula “statistically lower” instead of “significantly lower”, to avoid ambiguity.

The reviewer is right. We have changed the wording of the correlation analysis results to “positive (or negative) weak correlation” and we substituted “significantly lower” with “statistically lower” in the whole manuscript.

The small number of both extremely premature neonates and of adverse clinical outcomes may have contributed for some expected or apparent associations having reached no statistical significance; it should be acknowledges in the Discussion.

We agree with the reviewer and have acknowledged this fact in the discussion (Line 252) and limitations (Line 350).

From my point of view, Figure 2 does not add anything to Table 2.; Figure 1 requires better explanation of the content of the graphic and the graphics in Figure 3 and 4 might improve adding regression lines (ou curves).

We have deleted Figure 2, described the content of Figure 1 (now Figure 3 in the revised manuscript) in more detail (Line 207-214), and added regression lines to the scatter plots (now Figures 1 and 2).

Some minor comments and proposals of corrections are to be found in the annexed document but I would like to highlight that there is no evidence of a rationale to use "randomized controlled trials" to assess an observation of physiologic thresholds (line 40).

We would like to thank Reviewer 2 for the further comments and suggestions for minor changes. The sentence about randomized controlled trials in the introductions has been removed.

As the two other reviewers raised concerns regarding clarity of goals and objective, we decided to focus on the associations between body temperature and NIRS parameters as our primary outcome, and to report on already well known differences in body temperature between term and preterm infants as secondary outcome. We have thoroughly revised the whole manuscript and hope that the manuscript has improved in clarity of goals and objective.

Thank you for the valuable and supportive review!

Reviewer 3 Report

I would like thank authors for attempting to address a question which is often overlooked but may hold very important space in describing normal or abnormal changes in physiological signs at the time of birth. Overall this is well written. My critique is as following.

The major problem with this study is lack of clarity of goals and objective. Title seems to imply that this study is about continuous temperature/oxygen changes from fetal to neonatal transition but everything about the manuscript is about only one temperature event at minute 15 of life. It remains unclear to me why authors chose minute 15 as the point of interest. While there are good number of studies showing correlation of admission temperature and morbidity.

Specifics

Introduction seems unnecessarily too long; authors should assume the readers of this manuscript know the problems associated with hypo/hyperthermia with health of newborns.

I would advise authors to re-write introduction as there is awkward transition to objectives. I would advise authors to summarize their objectives in bullets types headings.

Redundancy in tables and figures. I would suggest that authors choose one method of showcasing their data.

Discussion needs to improve considerably. Authors spend a lot of time quoting previous studies to conclude that their study had lower incidence of hypothermia compared to what has previously been reported. They have attempted to answer that by arguing being mono-centric study and extra care of newborn. One other reason could just be by the fact that population included here was part of previously conducted five different studies.  We know that being part of a study in itself improves care. The positive correlation between temperature and heart rate is a well-known physiological phenomenon. I would like authors to bring that in their discussion as a possible reason and quote those studies by Davies P et al and others.

I was more interested in what this study did not find. This study does not show any correlation with hypothermia or hyperthermia and its effects on other vital signs including HR, SpO2, cerebral or peripheral NIRS saturations. Only a negative correlation of prSO2 in preterm. I would like authors to comment on this aspect of the study more on reasons behind these findings and quote more references to cement the argument that autoregulation may be more effective even in preterm infants.

Methodology section needs more clarity.

  1. In exclusion criteria – I suggest add all vaginal deliveries
  2. Under post-natal stabilization, were the infants placed on servo mode for auto regulation of overhead temperature in warmer? or was heat manually regulated?
  3. Results line 167- VPI and LPI had lowest temperature compared to which group? that seem unclear. I think you try to address this in line 170-173. I think these can be consolidated into one sentence.
  4. Can authors please comment on how much data was rejected if the data did not meet predefined statistical criteria of SpO2 being equal or below the regional saturation. As reported in one study on use of NIRS that it can be up to 50%. Vesoulis et al 2019. What effect does this have on validity of this study.

  1. I would like authors to comment on whether they can present data on delayed cord clamping in the subject population as we know extra blood received during delayed cord clamping has potential to improve cerebral perfusion.

  1. It may very well be that after birth the initial resuscitation has more effect on regional saturations with NIRS than does body temperature. I would like author’s comment or show data on effects of respiratory support including percentage of inspired oxygen on regional saturations if possible.

  1. Table 1 – I would suggest you add meaning of parentheses – median with interquartile range in the legend.

Author Response

I would like thank authors for attempting to address a question which is often overlooked but may hold very important space in describing normal or abnormal changes in physiological signs at the time of birth. Overall this is well written. My critique is as following.

The major problem with this study is lack of clarity of goals and objective. Title seems to imply that this study is about continuous temperature/oxygen changes from fetal to neonatal transition but everything about the manuscript is about only one temperature event at minute 15 of life. It remains unclear to me why authors chose minute 15 as the point of interest. While there are good number of studies showing correlation of admission temperature and morbidity.

Specifics

Introduction seems unnecessarily too long; authors should assume the readers of this manuscript know the problems associated with hypo/hyperthermia with health of newborns. I would advise authors to re-write introduction as there is awkward transition to objectives. I would advise authors to summarize their objectives in bullets types headings.

The reviewer is right. We have revised and shortened the introduction and re-structured the objectives.

Redundancy in tables and figures. I would suggest that authors choose one method of showcasing their data.

We have deleted Figure 2 as it does not add anything to Table 2. However, we think that Figure 1 (Boxplot) and Figures 3 and 4 (Figures 1 and 2 in the revised manuscript) illustrate our data in a comprehensive way and would prefer to keep them.

Discussion needs to improve considerably. Authors spend a lot of time quoting previous studies to conclude that their study had lower incidence of hypothermia compared to what has previously been reported. They have attempted to answer that by arguing being mono-centric study and extra care of newborn. One other reason could just be by the fact that population included here was part of previously conducted five different studies.  We know that being part of a study in itself improves care.

We agree with the reviewer. We have shortened the discussion distinctly and tried to focus more on the primary outcome parameters, i.e. the association between tissue oxygen saturation and body temperature. In the limitations section we already addressed that being included in a study might have biased the body temperature management: “Finally, a bias might be that all of the included neonates participated in clinical studies and as a result more attention may have been paid to body temperature management and preservation.” (Line 352)

The positive correlation between temperature and heart rate is a well-known physiological phenomenon. I would like authors to bring that in their discussion as a possible reason and quote those studies by Davies P et al and others.

Thank you for the input. We have added this to our discussion and references (Line 269).

I was more interested in what this study did not find. This study does not show any correlation with hypothermia or hyperthermia and its effects on other vital signs including HR, SpO2, cerebral or peripheral NIRS saturations. Only a negative correlation of prSO2 in preterm. I would like authors to comment on this aspect of the study more on reasons behind these findings and quote more references to cement the argument that autoregulation may be more effective even in preterm infants.

These concerns are in concordance with the other reviewers. Therefore, we decided to focus on the associations between body temperature and NIRS parameters as our primary outcome. We have thoroughly revised the whole manuscript and added several potential factors that may explain and/or have influenced our findings, among them cerebral autoregulation, mode of delivery and timing of umbilical cord clamping, to the discussion.

Methodology section needs more clarity.

  1. In exclusion criteria – I suggest add all vaginal deliveries
    We have added vaginally delivered newborns as an exclusion criterion.

  2. Under post-natal stabilization, were the infants placed on servo mode for auto regulation of overhead temperature in warmer? or was heat manually regulated?
    We agree that this important information is missing. The overhead temperature heat was manually regulated. We have added this information to the “Postnatal stabilization and temperature management” section (Line 76).

  3. Results line 167- VPI and LPI had lowest temperature compared to which group? that seem unclear. I think you try to address this in line 170-173. I think these can be consolidated into one sentence.
    In the version for peer-review, line 167 is the legend of Table 1.
    Lines 170-173: “EPI had the highest body temperature (median [IQR] 37.0 [36.9-37.4]°C) and VPI (median [IQR] 36.7 [36.3-37.0]°C, p=0.039) and LPI (median [IQR] 36.7 [36.4-37.0]°C, p=0.017) had the lowest mean body temperature compared to the other groups 15 minutes after birth (Figure 3).”

Hence, VPI and LPI had the same median body temperature, so they had the lowest body temperature compared to all the other groups. We added “compared to the other groups” to clarify.

  1. Can authors please comment on how much data was rejected if the data did not meet predefined statistical criteria of SpO2 being equal or below the regional saturation. As reported in one study on use of NIRS that it can be up to 50%. Vesoulis et al 2019. What effect does this have on validity of this study.

This was a retrospective study and we extracted data from our database. We always screen and clean our data (according to the described quality criteria) before adding it to the database. Therefore, we cannot answer the question of how much data was rejected. This is our standard procedure how to process with the data, which is an advantage of all of our NIRS studies, as we can always report on very clean and physiologically valid data.

  1. I would like authors to comment on whether they can present data on delayed cord clamping in the subject population as we know extra blood received during delayed cord clamping has potential to improve cerebral perfusion.

All included infants received cord clamping within 30 seconds after birth. Unfortunately we do not have data of infants with delayed cord clamping in the subject population, given the inclusion criteria of the five prospective observational studies.

  1. It may very well be that after birth the initial resuscitation has more effect on regional saturations with NIRS than does body temperature. I would like author’s comment or show data on effects of respiratory support including percentage of inspired oxygen on regional saturations if possible.

We fully agree with the reviewer’s assumption. This is one limitation of this study, as already mentioned in the limitations section: “Thirdly, we did only record the need for respiratory support, but data about the fraction of inspired oxygen were not available in our database.” (Line 347)

  1. Table 1 – I would suggest you add meaning of parentheses – median with interquartile range in the legend.

Correction done.

Thank you for the valuable and supportive review!

Round 2

Reviewer 1 Report

Dear Authors,

It is a pleasure to evaluate the revised version of the manuscript “Association between regional tissue oxygenation and body temperature in term and preterm infants born by Caesarean section”. As said in the prior analysis, the study is highly original, interesting, and important. The authors modified substantially the manuscript, and, in my opinion, they addressed the main criticisms and suggestions made in the prior version.   There is only one minor issue that remained in the text:  

Minor issues:

  • Line 34: “However, the literature about normal body temperature is inconsistent and the normal range recommended by the WHO is under discussion[2][3].” – The references are 15-year old and the “nowadays” seems out of place.

In conclusion, the revision was excellent and it addressed the main concerns raised in the previous version.

Author Response

Dear Reviewer,

Thank you, we deleted the sentence and thank your for the for the valuable and supportive review!

Reviewer 3 Report

Thank you Authors for addressing all comments. 

Author Response

Thank you for the valuable and supportive review!